# Training-Free Token Pruning via Zeroth-Order Gradient Estimation in Vision-Language Models

## Abstract

Large Vision–Language Models (VLMs) enable strong multimodal reasoning but incur heavy inference costs from redundant visual tokens. Token pruning alleviates this issue, yet existing approaches face limitations. Attention-based methods rely on raw attention scores, which are often unstable across layers and heads and can lead to redundant selections. Diversity-based methods improve robustness by selecting tokens far apart in feature space but risk dropping regions needed for accurate prediction. We propose *ZOO-Prune*, a training-free framework built on a simple intuition: tokens with higher sensitivity are more likely to influence the model's output, and they should also capture complementary visual cues rather than overlapping information. To achieve this, we estimate token sensitivity using zeroth-order perturbations at the projection layer, a shallow and computationally light component of the model. This approach measures how small random perturbations affect the projection outputs, allowing us to approximate each token's influence through lightweight forward passes without backpropagation. Extensive experiments across multiple VLMs and benchmarks show that *ZOO-Prune* consistently outperforms prior methods, pruning up to 94.4% of tokens while maintaining accuracy and significantly improving efficiency, achieving up to 2.30× faster end-to-end inference over the baseline.

## 1 Introduction

Large Vision–Language Models (VLMs) (Bai et al., 2023; Liu et al., 2024b; Zhu et al., 2025) achieve strong multimodal understanding but at a substantial computational cost. A major contributor to this cost is the large number of visual tokens produced by modern vision backbones. For example, the vision encoder in LLaVA-1.5 (Liu et al., 2024a) generates up to 576 tokens for a single image, whereas the text side often contains only a few tokens, such as *"Describe this image in a short sentence"*. This imbalance leads to high inference latency and memory overhead (Chen et al., 2024a; Alvar et al., 2025). To address this issue, token pruning has emerged as a practical solution that selectively removes less informative visual tokens at inference time. Prior work has shown that aggressive token reduction can yield large speedups with only modest drops in accuracy (Zhang et al., 2025; Yang et al., 2025).

Recent research on VLM token pruning has increasingly focused on training-free schemes. Unlike methods that require calibration data or fine-tuning (Hudson & Manning, 2019; Li et al., 2025; Lin et al., 2025), training-free approaches prune tokens directly at inference. These methods can be broadly categorized into two groups. (1) Attention-based approaches score tokens using attention magnitudes; however, attention often concentrates on background regions (Darcet et al., 2023) and tends to retain redundant tokens with overlapping content (Yang et al., 2025; Shang et al., 2024). For example, in an image of a laptop on a desk (Fig. 1), attention-based pruning may preserve many redundant tokens with overlapping content from the background, while ignoring tokens near the monitor that are critical for answering a question about objects on the monitor. (2) Diversity-based approaches (Alvar et al., 2025) select tokens by maximizing feature diversity, measuring pairwise distances between token embeddings. However, because these methods prioritize diversity uniformly across all tokens without explicitly considering task-relevant cues, they may discard tokens from visually salient regions. In the same laptop example, diversity-based pruning may select

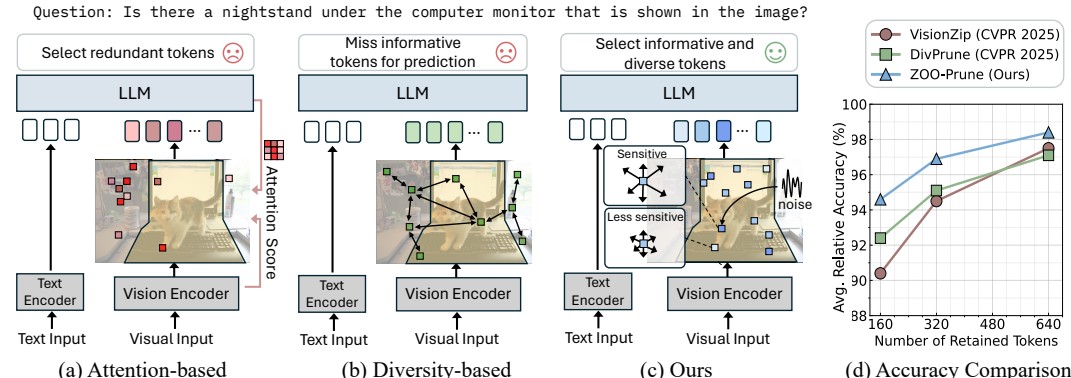

Figure 1: Illustration of training-free VLM token pruning methods. (a) Attention-based methods select tokens using attention scores, but often retain redundant tokens. (b) Diversity-based methods select tokens with different features to maximize coverage but may lose tokens located in semantically relevant regions (*e.g.*, around the monitor, highlighted in yellow). (c) Our method employs zeroth-order gradient estimation to quantify token sensitivity and integrates these scores into a diversity objective. (d) Accuracy comparison with LLaVA-NeXT-7B across 9 benchmarks, showing that ours outperforms both VisionZip (attention-based) and DivPrune (diversity-based) methods.

feature-diverse tokens, but not necessarily those near the monitor, which are crucial for reasoning about nearby objects.

Given these limitations, selecting a smaller subset of tokens while preserving visual information without degrading task performance remains a challenging problem. To tackle this, we investigate a novel metric for scoring each token, explicitly considering token sensitivity. Directly computing token sensitivities via gradients is costly, especially for VLM, motivating the use of zeroth-order gradient estimation (Nesterov & Spokoiny, 2017). This method quantifies how perturbations in input tokens influence model outputs using only forward passes, avoiding backpropagation. However, a naive application of zeroth-order estimation would require additional forward passes through the vision encoder, incurring substantial computational overhead. We therefore empirically examined whether sensitivity rankings obtained at a lightweight intermediate stage could serve as a reliable proxy for those derived from the full vision encoder. Intuitively, the projection layer is a natural choice since tokens here already capture high-level semantics from the vision encoder and are directly aligned with the language model. Our analysis revealed strong alignment between rankings computed from the vision encoder and those from the projection layer (Section 3.2).

Motivated by this finding, we propose a training-free VLM token pruning method named *ZOO-Prune (ZerOth-Order gradient estimation for token pruning)*. Our method measures each token's influence by injecting Gaussian noise at a lightweight projection layer and estimating the resulting gradient norms, which we define as sensitivity. To reduce redundancy while retaining informative tokens, we introduce Sensitivity-Aware Diversity Selection, which prioritizes tokens with high sensitivity and ensures sufficient feature diversity, inspired by (Alvar et al., 2025). By jointly considering sensitivity and diversity, *ZOO-Prune* produces pruned token subsets that preserve task-relevant information and enable effective compression even under aggressive pruning regimes (Fig. 1).

Our main contributions are as follows:

- We propose *ZOO-Prune*, a training-free pruning framework that unifies sensitivity and diversity, ensuring that pruned tokens are not only highly informative but also complementary, overcoming the limitations of importance-only or diversity-only methods.
- We introduce a zeroth-order sensitivity estimator at the projection layer, which provides stable token importance rankings with lightweight forward computations, eliminating the need for backpropagation or costly full-encoder passes.
- We demonstrate through extensive experiments on multiple VLMs and benchmarks that *ZOO-Prune* delivers superior accuracy–efficiency trade-offs, retaining up to 94.4% fewer tokens while maintaining accuracy and significantly reducing inference cost, achieving up to 2.30× faster end-to-end inference over the baseline.

## 2 RELATED WORK

### 2.1 VISION-LANGUAGE MODELS

Large Multimodal Models (LMMs), particularly Vision-Language Models (VLMs) (Liu et al., 2023; Dai et al., 2023; Chen et al., 2024b; Liu et al., 2024b; Bai et al., 2023), have demonstrated remarkable capabilities in multimodal reasoning and dialogue. Pioneering architectures like LLaVA (Liu et al., 2023) established a successful paradigm by aligning a pre-trained vision encoder (e.g., CLIP ViT (Radford et al., 2021)) with an instruction-tuned LLM through a simple projection layer. This design, further refined in subsequent works such as LLaVA family (Liu et al., 2024a;b), InternVL series (Chen et al., 2024b; Wang et al., 2024; Zhu et al., 2025; Wang et al., 2025), and Qwen-VL (Bai et al., 2023), enables strong visual understanding but introduces a significant computational challenge. These models typically encode a single image into hundreds of visual tokens, leading to substantial inference overhead. The computational burden is exacerbated when handling higher resolutions; for instance, LLaVA (Liu et al., 2023; 2024a;b) typically encodes $336 \times 336$ images into 576 tokens, and up to 2880 tokens at $672 \times 672$ resolution. The inherent redundancy within these extensive visual token sequences has motivated research into visual token pruning as a key approach for efficient VLMs inference.

### 2.2 VISUAL TOKEN PRUNING FOR VLMS

Visual token pruning methods are proposed to reduce the inference complexity of large VLMs by removing redundant visual representations. Existing pruning methods can be broadly divided into two categories: those that require fine-tuning or calibration, and those that are entirely training-free. The first category relies on additional data or adaptation to guide token reduction. CrossGET (Shi et al., 2024) and MADTP (Cao et al., 2024) introduce modality-specific tokens to align cross-modal features and drive token selection, while DeCo (Yao et al., 2024) employs adaptive pooling to decouple token compression from semantic abstraction at the patch level. VTW (Lin et al., 2025) removes all vision tokens after a specific layer, identified using a small calibration set and a KL-divergence criterion. FitPrune (Ye et al., 2025) reduces visual tokens in the multi-head attention of each layer via binary search, guided by attention statistics collected from inference or calibration examples. While effective, these methods require calibration data and model-specific adaptation, limiting their flexibility across architectures and deployment scenarios.

Training-free pruning avoids retraining and offers plug-and-play acceleration. Attention-based methods estimate token importance directly from attention magnitudes. FastV (Chen et al., 2024a), LLaVA-PruMerge (Shang et al., 2024), and VisionZip (Yang et al., 2025) remove or merge tokens based on early attention maps, while SparseVLM (Zhang et al., 2025) leverages question-driven cross-attention to induce dynamic sparsity. However, attention scores are often unstable and may retain semantically redundant tokens (Alvar et al., 2025; Lin et al., 2025), which limits performance under aggressive pruning. Another direction emphasizes feature diversity. DivPrune (Alvar et al., 2025) formulates token selection as a max-min diversity problem solved via greedy farthest-point sampling, reducing redundancy and preserving robustness at high pruning ratios. Yet by treating all tokens equally, it can overlook semantically critical regions and drop task-relevant information. To overcome these limitations, we propose a zeroth-order sensitivity estimator that quantifies each token's effect on the output without backpropagation. We further combine this signal with a diversity-based selection strategy, ensuring that pruning retains the most influential tokens while still covering complementary visual content, which leads to a stronger balance between efficiency and accuracy.

## 3 METHODOLOGY

### 3.1 BACKGROUND: ZEROTH-ORDER GRADIENT ESTIMATION

Zeroth-order (ZO) optimization provides a gradient-free alternative to first-order methods by using only forward queries of a function. It is particularly useful when exact gradient computation is infeasible, such as in black-box optimization (Sawada et al., 2025), adversarial attacks (Chen et al., 2017), or efficient fine-tuning of large models (Park et al., 2025; Zhang et al., 2024b). By relying solely on forward evaluations, ZO methods circumvent the need for backpropagation, thereby reducing memory costs and enabling applications to complex or non-differentiable modules.

A widely used estimator is the *randomized gradient estimator* (RGE) (Duchi et al., 2015; Nesterov & Spokoiny, 2017), which approximates gradients by finite differences along random directions. Given a function $f : \mathbb{R}^{d_1} \to \mathbb{R}^{d_2}$, the central-difference RGE with $m$ queries is

$$\widehat{\nabla} f(x) \;=\; \frac{1}{m} \sum_{j=1}^{m} \frac{f(x + hu_j) - f(x - hu_j)}{2h} \, u_j, \quad u_j \sim \mathcal{N}(0, I_d), \tag{1}$$

where $h > 0$ is a small step size.

### 3.2 OBSERVATION: PROJECTOR AS A PROXY FOR END-TO-END VISION SENSITIVITY

Original zeroth-order methods aim to reconstruct the full gradient vector, but in our setting, we only require a *relative ranking* of token importance. A naive approach would apply RGE directly to the vision encoder, incurring substantial computational overhead due to additional end-to-end forward passes. For instance, suppose an image is tokenized into $n = 500$ visual tokens with $m = 64$ random perturbation directions per token. Since RGE requires two forward passes per direction, the total cost scales as $2nm$ forward passes, amounting to $\sim 6.4 \times 10^6$ GFLOPs, which is clearly prohibitive.

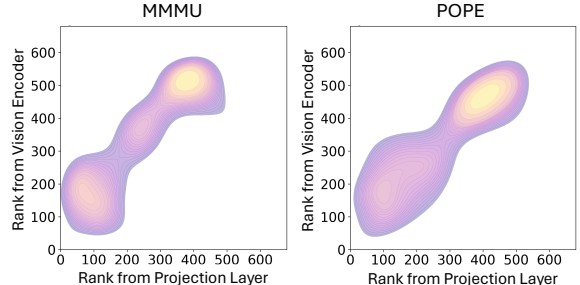

Figure 2: Kernel density estimate (KDE) of Spearman rank correlations between token-importance rankings from the *Vision encoder* and the *Projection layer* on the MMMU and POPE datasets. Each dataset shows Spearman correlation of 0.55 and 0.49, respectively. Detailed setting is described in Appendix A.

To reduce this cost, we empirically evaluated whether sensitivity rankings obtained at a lightweight intermediate stage (*i.e.*, projection layer) could serve as a reliable proxy for those derived from the full vision encoder. In Fig. 2, we rank token sensitivities based on RGE and visualize the Spearman rank correlations across two datasets. Our analysis shows a strong alignment between token importance rankings computed from the vision encoder outputs and those obtained from the projection layer. Beyond empirical evidence, the projection layer can be seen as a modality-aligning bottleneck. It consolidates high-level semantic information from the vision encoder and maps it into the language embedding space, naturally emphasizing tokens that are important for downstream predictions. Since token pruning only requires relative importance rather than exact gradients, this layer provides a compact, semantically meaningful proxy that preserves token sensitivity rankings.

Motivated by this finding, we compute token sensitivities at the projection layer, which contains only a few layers and therefore introduces negligible additional cost during inference. By relying solely on forward queries, this approach avoids the expensive backpropagation required by first-order methods, making zeroth-order sensitivity estimation a practical and efficient tool for token-level analysis in large-scale VLMs.

### 3.3 ZOO-PRUNE: ZEROTH-ORDER GRADIENT ESTIMATION FOR TOKEN PRUNING

Fig. 3 illustrates the overall *ZOO-Prune* framework. Given the outputs of the vision encoder, our method first computes token-level sensitivity using zeroth-order gradient estimation and then uses these scores in a diversity-aware selection procedure to produce a subset of tokens that is passed to the LLM. Mathematically, let $X \in \mathbb{R}^{N_v \times d_v}$ denote $N_v$ vision tokens with $d_v$ dimension, and

$$Z = M(X) \in \mathbb{R}^{N_v \times d_l}$$

the projected embeddings through the multimodal projection layer $M : \mathbb{R}^{d_v} \to \mathbb{R}^{d_l}$. We sample $m$ random perturbation directions $\{u_j\}_{j=1}^{m}$, where $u_j \sim \mathcal{N}(0, I_{d_v})$ normalized to unit norm. For each token $i$, we measure the symmetric finite-difference response:

$$\delta_{i,j} \;=\; \frac{M(x_i + hu_j) - M(x_i - hu_j)}{2h}, \tag{2}$$

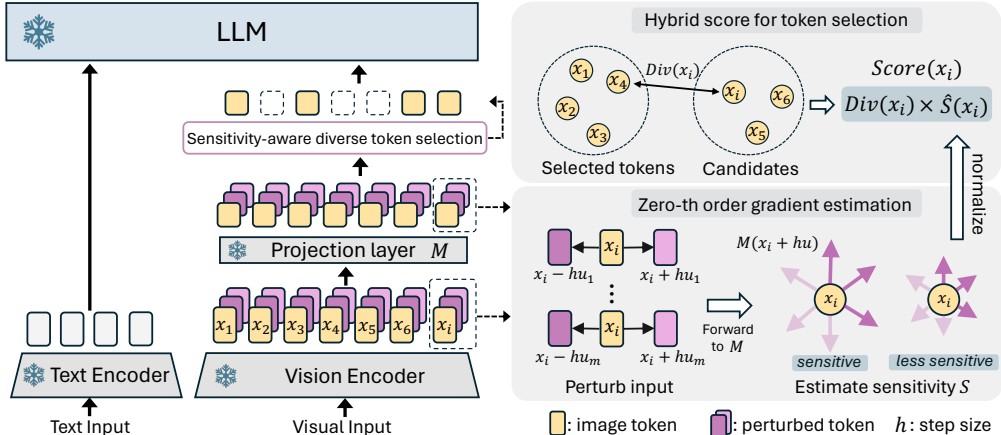

Figure 3: Overview of *ZOO-Prune*. Given visual tokens from the vision encoder, we estimate token sensitivity via zeroth-order gradient approximation at the projection layer by adding Gaussian perturbations (*i.e.*, $x_i \pm h u_j$). The resulting sensitivity scores are integrated with a diversity objective to form a hybrid score, guiding the selection. The selected subset is then passed to the LLM together with the text input, enabling efficient multimodal reasoning with reduced computation.

where $x_i \in \mathbb{R}^{d_v}$ is the $i$-th vision token. We then define the *token sensitivity* of token $i$ as the average response magnitude:

$$S(i) \;=\; \frac{1}{m} \sum_{j=1}^{m} \|\delta_{i,j}\|_2. \tag{3}$$

This metric estimates the *approximated mean sensitivity* of each token shown in Proposition 3.1.

**Proposition 3.1** (Approximated Mean Sensitivity). *Let $M : \mathbb{R}^n \to \mathbb{R}^m$ be differentiable at $x \in \mathbb{R}^n$ with Jacobian $J(x) = \nabla M(x)$. Let $u \sim \mathcal{N}(0, I_n)$ be an isotropic Gaussian perturbation and $h > 0$ a small step size. Define the finite-difference sensitivity $S(x) = \mathbb{E}_u \left[ \left\| \frac{M(x+hu)-M(x-hu)}{2h} \right\|_2 \right]$. Then, for sufficiently small $h$,*

$$S(x) = \mathbb{E}_u \big[ \|J(x)u\|_2 \big] + O(h^2).$$

The detailed proof is provided in Appendix B. The proposition shows that, unlike traditional RGE, which estimates gradient direction, our sensitivity metric captures the magnitude of local response. Specifically, $\mathbb{E}_u[\|J(x)u\|_2]$ quantifies how much the output changes on average under random perturbations, providing a scalar measure of token influence.

**Sensitivity-aware diversity selection.** While sensitivity captures the most informative tokens, it does not by itself enforce coverage over diverse visual content. To reduce redundancy, we integrate a diversity criterion inspired by DivPrune (Alvar et al., 2025). Let $Z_i$ denote the vision feature of token $i$, and $\mathcal{P}$ the set of already selected tokens. We define

$$\text{Div}(i, \mathcal{P}) = 1 - \max_{j \in \mathcal{P}} \cos(Z_i, Z_j), \tag{4}$$

where $\cos(\cdot, \cdot)$ is cosine similarity. The final selection score is defined as

$$\text{Score}(i) = \widehat{S}(i) \cdot \text{Div}(i, \mathcal{P}), \tag{5}$$

where $\widehat{S}(i)$ is the normalized sensitivity score. The multiplicative design avoids introducing additional hyperparameters for weighting the two criteria. Compared to DivPrune, our diversity selection method introduces two key modifications: (1) For the first token, DivPrune selects the one that is maximally distant from all others, whereas ours prioritizes the token with the highest sensitivity. (2) For subsequent selections, DivPrune considers only diversity, while our method combines sensitivity and diversity via the hybrid score above. This procedure yields a token subset that is both sensitivity-driven and diversity-driven. The overall process is described in Algorithm 1.

---

**Algorithm 1** *ZOO-Prune*: ZerOth-Order Sensitivity-Aware Token Pruning

---

1: **Input:** Vision tokens $X \in \mathbb{R}^{N_v \times d_v}$, projection $M$, number of tokens to select $k$, step size $h$, number of perturbations $m$
2: **Output:** Selected token indices $\mathcal{P}$
3: % —— [ZOO-based Sensitivity Estimation] ———-
4: Sample $m$ random perturbations $U \in \mathbb{R}^{m \times d_v}$, $u_j \sim \mathcal{N}(0, I)$, normalized to $\|u_j\|_2 = 1$
5: Expand $X$ along perturbations: $X^+ = X + hU$, $X^- = X - hU$
6: Project perturbed features: $Z^+ = M(X^+)$, $Z^- = M(X^-)$
7: Compute finite-difference responses: $\Delta = \frac{Z^+ - Z^-}{2h}$
8: Sensitivity: $S(i) = \frac{1}{m} \sum_{j=1}^{m} \|\Delta_{i,j}\|_2$
9: % —— [Sensitivity-Aware Diversity Selection ] ——
10: Normalize sensitivities: $\widehat{S}(i) = \frac{S(i) - \min_j S(j)}{\max_j S(j) - \min_j S(j)}$
11: Initialize $\mathcal{P} \leftarrow \emptyset$
12: **while** $|\mathcal{P}| < k$ **do**
13:     Compute diversity: $\text{Div}(i, \mathcal{P}) = 1 - \max_{j \in \mathcal{P}} \cos(Z_i, Z_j)$    (set to 1 if $\mathcal{P}$ is empty)
14:     Fusion score: $\text{Score}(i) = \widehat{S}(i) \cdot \text{Div}(i, \mathcal{P})$
15:     Select $i^\star = \arg\max_i \text{Score}(i)$
16:     $\mathcal{P} \leftarrow \mathcal{P} \cup \{i^\star\}$
17: **end while**
18: **return** $\mathcal{P}$

---

Table 1: Performance Comparison on LLaVA-1.5-7B.

| Method | GQA Acc. ↑ | MMB Acc. ↑ | MME P+C ↑ | POPE F1 ↑ | SQA Acc. ↑ | VQA$^{V2}$ Acc. ↑ | VQA$^{Text}$ Acc. ↑ | MMMU Acc. ↑ | SEED Acc. ↑ | Avg. ↑ |
|---|---|---|---|---|---|---|---|---|---|---|
| *Total 576 Tokens* | | | | | | | | | | |
| LLaVA-1.5-7B | 61.90 100% | 64.70 100% | 1862.00 100% | 85.90 100% | 69.50 100% | 78.50 100% | 58.20 100% | 36.30 100% | 58.60 100% | 100% |
| *Retain 192 Tokens ↓ 66.7%* | | | | | | | | | | |
| FastV (ECCV 2024) | 52.70 | 61.20 | 1612.00 | 64.80 | 67.30 | 67.10 | 52.50 | 34.30 | 57.10 | 89.6% |
| SparseVLM (ICML 2025) | 57.60 | 62.50 | 1721.00 | 83.60 | 69.10 | 75.60 | 56.10 | 33.80 | 55.80 | 95.5% |
| VisionZip (CVPR 2025) | 59.30 | **63.00** | **1782.60** | 85.30 | 68.90 | 76.80 | 57.30 | 36.60 | 56.40 | 97.9% |
| DivPrune (CVPR 2025) | 59.97 | 62.54 | 1762.23 | 87.00 | 68.91 | 76.87 | 56.97 | 35.44 | 58.71 | 98.0% |
| ***ZOO-Prune* (Ours)** | **60.03** | 62.89 | 1781.66 | **87.24** | **69.16** | 77.34 | 57.30 | 36.11 | 58.80 | 98.6% |
| *Retain 128 Tokens ↓ 77.8%* | | | | | | | | | | |
| FastV (ECCV 2024) | 49.60 | 56.10 | 1490.00 | 59.60 | 60.20 | 61.80 | 50.60 | 34.90 | 55.90 | 84.5% |
| SparseVLM (ICML 2025) | 56.00 | 60.00 | 1696.00 | 80.50 | 67.10 | 73.80 | 54.90 | 33.80 | 53.40 | 93.0% |
| VisionZip (CVPR 2025) | 57.60 | 62.00 | **1761.70** | 83.20 | 68.90 | 75.60 | 56.80 | **37.90** | 54.90 | 96.8% |
| DivPrune (CVPR 2025) | 59.25 | **62.03** | 1718.22 | 86.72 | **68.96** | 75.96 | 56.06 | 35.56 | 56.98 | 96.9% |
| ***ZOO-Prune* (Ours)** | **59.49** | 61.86 | 1751.60 | **87.13** | 68.91 | 76.57 | 57.87 | 35.67 | **57.53** | 97.8% |
| *Retain 64 Tokens ↓ 88.9%* | | | | | | | | | | |
| FastV (ECCV 2024) | 46.10 | 48.00 | 1256.00 | 48.00 | 51.10 | 55.00 | 47.80 | 34.00 | 51.90 | 75.5% |
| SparseVLM (ICML 2025) | 52.70 | 56.20 | 1505.00 | 75.10 | 62.20 | 68.20 | 51.80 | 32.70 | 51.10 | 87.0% |
| VisionZip (CVPR 2025) | 55.10 | 60.10 | **1690.00** | 77.00 | **69.00** | 72.40 | **55.50** | 36.20 | 52.20 | 93.1% |
| DivPrune (CVPR 2025) | 57.78 | 59.28 | 1674.40 | 85.56 | 68.17 | 74.11 | 54.69 | **35.56** | 55.13 | 94.8% |
| ***ZOO-Prune* (Ours)** | **58.47** | **60.22** | 1675.59 | **85.86** | 68.27 | 75.02 | 55.35 | 35.44 | **55.84** | 95.5% |

## 4 EXPERIMENTS

We evaluate *ZOO-Prune* on LLaVA-v1.5-7B/13B (Liu et al., 2024a), LLaVA-1.6-7B (Liu et al., 2024b), and Qwen2.5-VL-7B (Bai et al., 2025). LLaVA-v1.5 employs a CLIP ViT-L (Radford et al., 2021) with 576 tokens, LLaVA-NeXT scales to 2880 tokens for high-resolution inputs, and Qwen2.5-VL adopts a dynamic-resolution ViT encoder. Following VisionZip (Yang et al., 2025), we evaluate multiple pruning ratios and report performance relative to the unpruned baseline across nine benchmarks. For LLaVA-NeXT, we applied a low-rank factorization ($k = 128$) to the MM-projector layers to further boost efficiency, since this model processes a large number of visual tokens. All experiments are training-free and calibration-free, run on 4×A6000 GPUs with $m$=64, $h$=0.01, and evaluated using `lmms-eval` (Zhang et al., 2024a). More details in Appendix C.

Table 2: Performance Comparison on LLaVA-1.5-13B.

| Method | GQA Acc. ↑ | MMB Acc. ↑ | MME P+C ↑ | POPE F1 ↑ | SQA Acc. ↑ | $VQA^{V2}$ Acc. ↑ | $VQA^{Text}$ Acc. ↑ | MMMU Acc. ↑ | SEED-I Acc. ↑ | Avg. ↑ |
|---|---|---|---|---|---|---|---|---|---|---|
| *Total 576 Tokens* | | | | | | | | | | |
| LLaVA-1.5-13B | 63.20 100% | 67.70 100% | 1818.00 100% | 85.90 100% | 72.80 100% | 80.00 100% | 61.30 100% | 36.40 100% | 66.90 100% | 100% |
| *Retain 192 Tokens ↓ 66.7%* | | | | | | | | | | |
| VisionZip (CVPR 2025) | 59.10 | **66.90** | 1754.00 | 85.10 | **73.50** | 78.10 | **59.50** | 36.40 | 65.20 | 97.9% |
| DivPrune (CVPR 2025) | 59.42 | 66.58 | **1781.50** | 86.76 | 72.88 | 77.98 | 58.46 | 36.56 | **65.72** | 98.1% |
| *ZOO-Prune* (Ours) | **59.95** | 66.67 | 1762.41 | 86.73 | 73.12 | **78.65** | 59.11 | **37.33** | 65.56 | **98.6%** |
| *Retain 128 Tokens ↓ 77.8%* | | | | | | | | | | |
| VisionZip (CVPR 2025) | 57.90 | 66.70 | 1743.00 | 85.20 | **74.00** | 76.80 | 58.70 | **36.10** | 63.80 | 97.0% |
| DivPrune (CVPR 2025) | 58.89 | 66.07 | 1748.56 | 86.53 | 72.83 | 77.10 | 58.17 | 35.56 | 64.22 | 97.0% |
| *ZOO-Prune* (Ours) | **58.89** | **67.01** | **1791.10** | **86.95** | 73.38 | **77.83** | **58.80** | 35.56 | **64.50** | **97.8%** |
| *Retain 64 Tokens ↓ 88.9%* | | | | | | | | | | |
| VisionZip (CVPR 2025) | 56.20 | **64.90** | 1676.00 | 76.00 | **74.40** | 73.70 | 57.40 | **36.40** | 60.40 | 93.7% |
| DivPrune (CVPR 2025) | 57.66 | 64.60 | **1777.93** | 84.80 | 71.34 | 75.20 | 57.11 | 35.22 | 62.44 | 95.4% |
| *ZOO-Prune* (Ours) | **58.58** | 64.78 | **1780.03** | **85.34** | 72.09 | **76.39** | **58.59** | 36.00 | **63.02** | **96.5%** |

Table 3: Performance Comparison on LLaVA-NeXT-7B.

| Method | GQA Acc. ↑ | MMB Acc. ↑ | MME P+C ↑ | POPE F1 ↑ | SQA Acc. ↑ | $VQA^{V2}$ Acc. ↑ | $VQA^{Text}$ Acc. ↑ | MMMU Acc. ↑ | SEED-I Acc. ↑ | Avg. ↑ |
|---|---|---|---|---|---|---|---|---|---|---|
| *Total 2880 Tokens* | | | | | | | | | | |
| LLaVA-NeXT-7B | 64.20 100% | 67.90 100% | 1842.00 100% | 86.40 100% | 70.20 100% | 80.10 100% | 61.30 100% | 35.10 100% | 70.20 100% | 100% |
| *Retain 640 Tokens ↓ 77.8%* | | | | | | | | | | |
| SparseVLM (ICML 2025) | 60.30 | 65.70 | 1772.00 | – | 67.70 | 77.10 | 57.80 | 34.60 | – | 97.5% |
| VisionZip (CVPR 2025) | 61.30 | **66.30** | 1787.00 | 86.30 | **68.10** | 79.10 | **60.20** | 34.70 | 66.70 | 97.5% |
| DivPrune (CVPR 2025) | 61.58 | 65.38 | 1773.04 | 85.51 | 67.82 | 78.94 | 55.41 | 36.89 | 67.56 | 97.1% |
| *ZOO-Prune* (Ours) | **62.19** | 65.21 | **1816.45** | **86.75** | 68.02 | **79.64** | 57.98 | **36.89** | **67.95** | **98.3%** |
| *Retain 320 Tokens ↓ 88.9%* | | | | | | | | | | |
| SparseVLM (ICML 2025) | 57.70 | 64.30 | 1694.00 | – | 67.30 | 73.40 | 55.90 | 34.40 | – | 94.5% |
| VisionZip (CVPR 2025) | 59.30 | 63.10 | 1702.00 | 82.10 | 67.30 | 76.20 | **58.90** | 35.30 | 63.40 | 95.1% |
| DivPrune (CVPR 2025) | 59.63 | 63.66 | 1731.04 | 83.47 | **67.82** | 76.64 | 53.84 | **37.11** | 65.35 | 95.1% |
| *ZOO-Prune* (Ours) | **60.97** | **64.86** | **1787.68** | **85.47** | 67.77 | **78.08** | 57.28 | 37.00 | **66.47** | **97.1%** |
| *Retain 160 Tokens ↓ 94.4%* | | | | | | | | | | |
| SparseVLM (ICML 2025) | 51.20 | 63.10 | 1542.00 | – | 67.50 | 66.30 | 46.40 | 32.80 | – | – |
| VisionZip (CVPR 2025) | 55.50 | 60.10 | 1630.00 | 74.80 | 68.30 | 71.40 | **56.20** | 36.10 | 58.30 | 90.4% |
| DivPrune (CVPR 2025) | 57.79 | 62.29 | 1658.25 | 79.36 | 68.02 | 73.92 | 52.42 | 36.44 | 62.54 | 92.4% |
| *ZOO-Prune* (Ours) | **59.93** | **64.18** | **1738.64** | **83.05** | **68.42** | **76.12** | 55.42 | **37.11** | **64.05** | **95.4%** |

## 4.1 COMPARISON ON DIVERSE TASKS

**Results on LLaVA-1.5-7B and 13B.** As shown in Tables 1 and 2, *ZOO-Prune* consistently outperforms the state-of-the-art training-free pruning methods, especially under aggressive compression. On LLaVA-1.5-7B, it preserves 95.5% performance with only 64 tokens, surpassing DivPrune (Alvar et al., 2025) (94.8%) and far exceeding attention-based methods such as FastV (Chen et al., 2024a), which retains three times more tokens but drops to 89.6%. On LLaVA-1.5-13B, *ZOO-Prune* achieves 96.5% with 64 tokens and outperforms DivPrune by 1.7%, confirming that combining sensitivity and diversity effectively preserves reasoning under extreme pruning.

**Results on LLaVA-NeXT-7B.** We further evaluate *ZOO-Prune* on LLaVA-NeXT-7B, which processes up to 2880 tokens. As reported in Table 3, *ZOO-Prune* maintains 98.3% performance when pruning 77.8% of tokens, outperforming VisionZip (97.5%) and nearly matching the baseline. Even with a 94.4% pruning (160 tokens), it can still maintain a 95.4% performance, demonstrating the scalability of sensitivity-aware diversity selection under high-resolution compression.

**Results on Qwen2.5-VL-7B.** To validate the generalizability of *ZOO-Prune*, we evaluated it on Qwen2.5-VL-7B, a distinct VLM variant beyond the LLaVA family with dynamic-resolution in-

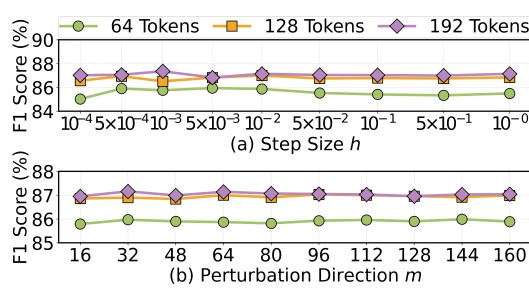

Figure 4: Hyperparameter sensitivity on POPE with LLaVA-1.5-7B: (a) effect of step size $h$, (b) effect of the number of perturbation directions $m$.

Table 4: Performance comparison on Qwen2.5-VL-7B with a dynamic resolution setting.

| Method | GQA | MMB | MME | POPE | Avg. |
|---|---|---|---|---|---|
| *Baseline (Full Tokens)* | | | | | |
| Qwen2.5-VL-7B | 60.84 | 84.10 | 2310 | 86.3 | 100% |
| *Retain 20% Tokens* | | | | | |
| VisionZip (CVPR 2025) | 57.27 | 79.72 | **2221** | 83.89 | 95.6% |
| DivPrune (CVPR 2025) | **60.05** | 79.55 | 2173 | 83.42 | 96.0% |
| ***ZOO-Prune* (Ours)** | 58.81 | **80.60** | 2201 | **84.17** | **96.2%** |
| *Retain 10% Tokens* | | | | | |
| VisionZip (CVPR 2025) | 54.09 | 76.03 | 1937 | 78.97 | 88.7% |
| DivPrune (CVPR 2025) | **55.49** | 76.03 | **2054** | 79.05 | 90.5% |
| ***ZOO-Prune* (Ours)** | 55.45 | **76.28** | 2018 | **80.99** | **90.8%** |

Table 5: Ablation on Token Selection Metrics with LLaVA-NeXT-7B.

| Sensitivity | Diversity | Fusion | GQA Acc. ↑ | MMB Acc. ↑ | MME P+C ↑ | POPE F1 ↑ | SQA Acc. ↑ | VQA$^{V2}$ Acc. ↑ | VQA$^{Text}$ Acc. ↑ | MMMU Acc. ↑ | SEED-I Acc. ↑ | Avg. ↑ |
|---|---|---|---|---|---|---|---|---|---|---|---|---|
| | | | *Retain 640 Tokens ↓ 77.8%* | | | | | | | | | |
| ✓ | | - | 61.23 | 65.21 | 1818.62 | 86.54 | 68.07 | 78.47 | 54.12 | 35.78 | 66.29 | 96.7% |
| | ✓ | - | 61.58 | 65.38 | 1773.04 | 85.51 | 67.82 | 78.94 | 55.41 | 36.89 | 67.56 | 97.1% |
| ✓ | ✓ | Sum | 61.81 | **65.55** | 1794.17 | 86.34 | **68.27** | 79.38 | **57.99** | **37.22** | 67.29 | 98.1% |
| ✓ | ✓ | Multiply | **62.19** | 65.21 | **1816.45** | **86.75** | 68.02 | **79.64** | 57.98 | 36.89 | **67.95** | **98.3%** |
| | | | *Retain 320 Tokens ↓ 88.9%* | | | | | | | | | |
| ✓ | | - | 59.22 | 64.69 | 1744.42 | 83.15 | 67.63 | 75.69 | 47.25 | 34.78 | 63.69 | 92.9% |
| | ✓ | - | 59.63 | 63.66 | 1731.04 | 83.47 | 67.82 | 76.64 | 53.84 | 37.11 | 65.35 | 95.1% |
| ✓ | ✓ | Sum | 60.47 | 64.43 | 1761.52 | 84.21 | **68.47** | 77.70 | 56.79 | 36.89 | 65.46 | 96.4% |
| ✓ | ✓ | Multiply | **60.97** | **64.86** | **1787.68** | **85.47** | 67.77 | **78.08** | 57.28 | **37.00** | **66.47** | **97.1%** |
| | | | *Retain 160 Tokens ↓ 94.4%* | | | | | | | | | |
| ✓ | | - | 57.23 | 61.86 | 1674.35 | 77.27 | 68.82 | 72.58 | 50.96 | 35.56 | 61.04 | 91.2% |
| | ✓ | - | 57.79 | 62.29 | 1658.25 | 79.36 | 68.02 | 73.92 | 52.42 | 36.44 | 62.54 | 92.4% |
| ✓ | ✓ | Sum | 59.48 | 63.49 | 1710.35 | 81.21 | 68.22 | 75.88 | 54.77 | 34.78 | 63.47 | 93.8% |
| ✓ | ✓ | Multiply | **59.93** | **64.18** | **1738.64** | **83.05** | 68.42 | **76.12** | **55.42** | **37.11** | **64.05** | **95.4%** |

puts. As shown in Table 4, *ZOO-Prune* continues to lead, achieving 96.2% performance with a 20% token budget and 90.8% with only 10% of the tokens, consistently surpassing both VisionZip and DivPrune. These results confirm that *ZOO-Prune* generalizes robustly across diverse VLM architectures, effectively preserving task performance even with long, variable-length token sequences.

## 4.2 ABLATION STUDIES AND ANALYSIS

**Ablation Studies.** We ablate four variants on LLaVA-NeXT-7B (Table 5): Sensitivity-only, Diversity-only (DivPrune), and two Fusion strategies (sum, multiply). Sensitivity-only selects task-critical tokens and excels on reasoning, but lags on TextVQA where context is vital. Diversity-only offers broad coverage yet misses key cues. Combining both is consistently better: Fusion (Multiply) delivers 98.3% at 22.2% tokens with no extra hyperparameters. These results show that sensitivity and diversity are complementary and jointly essential for training-free token pruning.

**Hyperparameter Analysis.** We analyze the sensitivity of *ZOO-Prune* to its two hyperparameters $m$ (number of perturbations) and $h$ (step size) on the POPE benchmark (Fig. 4). Performance is consistently stable across a wide range: accuracy remains highly consistent as $m$ varies from 16 to 160 and is similarly insensitive to $h$ over [1e-4, 1]. We adopt $m = 64$ to reduce variance at negligible cost, and fix $h = 0.01$ for all experiments, thereby eliminating the need for task-specific tuning.

**Inference Efficiency.** To assess efficiency, we measure end-to-end (E2E) latency and prefilling time in Fig. 5. Prefilling, dominated by visual token processing, is the main stage accelerated by pruning. All methods achieve notable gains, but *ZOO-Prune* consistently offers the best trade-off. At the most aggressive setting (160 tokens, 94.4% pruning), it reduces E2E latency by $2.30\times$ and prefilling by $2.59\times$. All experiments are conducted on a single NVIDIA L40S GPU. Despite a negligible overhead from sensitivity estimation, *ZOO-Prune* sustains markedly higher accuracy, establishing it as a practical solution for efficient VLM deployment. Additional analysis of FLOPs in Appendix D shows consistent trends with the latency and prefilling results.

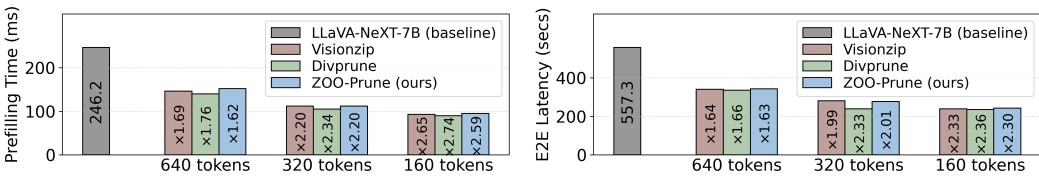

(a) Prefilling Time (ms)        (b) E2E Latency (secs)

Figure 5: Inference efficiency on the ScienceQA benchmark relative to the LLaVA-NeXT-7B baseline. The left panel reports prefilling time and the right panel shows end-to-end (E2E) latency. Speedup factors over the baseline are annotated inside each bar.

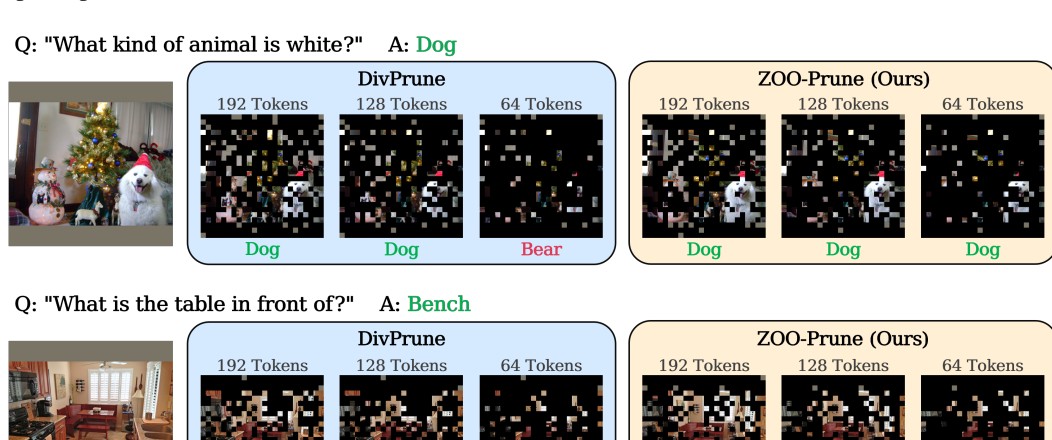

Figure 6: Qualitative comparison of pruned token masks from DivPrune and *ZOO-Prune* on the GQA benchmark under varying token budgets.

**Visualization.** Fig. 6 qualitatively compares *ZOO-Prune* with DivPrune under aggressive pruning on GQA. Although DivPrune preserves feature diversity, it often drops critical tokens, for example, omitting most dog tokens and predicting "bear," or focusing on a window instead of the queried object. In contrast, *ZOO-Prune* uses zeroth-order sensitivity to preserve key entities like the dog and bench. By combining sensitivity and diversity, it maintains correct predictions even at a 64-token budget, underscoring that sensitivity-aware diversity is essential for semantic integrity and reliable reasoning under heavy compression. See Appendix E for more examples.

## 5 CONCLUSION

In this paper, we presented *ZOO-Prune*, a training-free and attention-free token pruning framework for vision–language models that unifies zeroth-order sensitivity estimation with diversity-aware selection. By leveraging lightweight projection-layer responses, our approach efficiently measures token importance without backpropagation or additional end-to-end passes, while Sensitivity-Aware Diversity Selection ensures both informativeness and representational coverage. Extensive experiments across multiple benchmarks show that *ZOO-Prune* achieves state-of-the-art performance among training-free pruning methods, enabling substantial reductions in inference cost with minimal loss in accuracy. We believe this work highlights the potential of gradient-free sensitivity analysis for efficient multimodal learning and opens promising directions for scalable model compression.

**Limitations and Future Work.** While effective, *ZOO-Prune* currently relies on 32∼64 random perturbations per token for zeroth-order sensitivity estimation, which may increase overhead for very large token sets in future VLM architectures. It has also been validated primarily on existing VLMs such as LLaVA-NeXT, and its generalization to other multimodal models or modalities (*e.g.*, video, 3D) remains to be explored. A promising direction for future work is to investigate whether reliable token sensitivities can be estimated using only ∼4 perturbations, potentially further reducing computational cost.

**Reproducibility Statement.** Our method is detailed in Section 3 and Algorithm 1, with theoretical analysis in Appendix B.1. All experimental setup details, including models, benchmarks, and hyperparameters, are provided in Appendix C. Our work, built upon the open-source `lmms-eval` framework, will be released with full source code to ensure reproducibility.

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

# Appendix

This appendix supplements the main paper with the theoretical analysis of Proposition 3.1, additional details on the experimental configuration and evaluation strategy, and further qualitative results that illustrate the token preservation patterns across a range of scenarios. The contents are organized as follows:

- **Appendix A**: Implementation details for KDE experiments in Section 3.2.
- **Appendix B**: Theoretical analysis of the proposed sensitivity estimator.
- **Appendix C**: Experimental setup, implementation details, and evaluation protocols.
- **Appendix D**: Further Inference Efficiency Analysis: FLOPs.
- **Appendix E**: Additional qualitative results, including both successful and failure cases.
- **Appendix F**: The Use of Large Language Models (LLMs).

## A    EXPERIMENTAL DETAILS FOR SPEARMAN CORRELATION ANALYSIS

For the correlation analysis in Fig. 2, we measured the agreement between token-importance rankings derived from the vision encoder and the projection layer. Specifically, we selected 50 random samples per dataset from MMMU and POPE. For each sample, token sensitivities were first computed using RGE at both the vision encoder output and the projection layer. To ensure stable ranking comparisons, we applied a threshold of 0.5 to filter out low-sensitivity tokens before computing ranks. The Spearman's rank correlation coefficient was then calculated for each sample, and the distribution across 50 samples was visualized using a kernel density estimate (KDE) plot.

The resulting average Spearman correlations were 0.55 for MMMU and 0.49 for POPE, indicating a consistent alignment between token rankings obtained at the projection layer and those from the full vision encoder. This confirms that the projection layer can serve as a reliable proxy for token-level importance estimation while significantly reducing computational overhead.

## B    THEORETICAL ANALYSIS

### B.1    PROPOSITIONAL PROOF

**Proposition B.1** (Approximated Mean Sensitivity). *Let $M : \mathbb{R}^n \to \mathbb{R}^m$ be differentiable at $x \in \mathbb{R}^n$ with Jacobian $J(x) = \nabla M(x)$. Let $u \sim \mathcal{N}(0, I_n)$ be an isotropic Gaussian perturbation and $h > 0$ a small step size. Define the finite-difference sensitivity $S(x) = \mathbb{E}_u \left[ \left\| \frac{M(x+hu) - M(x-hu)}{2h} \right\|_2 \right]$. Then, for sufficiently small $h$,*

$$S(x) = \mathbb{E}_u \big[ \|J(x)u\|_2 \big] + O(h^2).$$

*Proof.* Since $M$ is differentiable at $x$, we apply a first-order Taylor expansion around $x$ for perturbations $hu$:

$$M(x + hu) = M(x) + hJ(x)u + O(h^2), \tag{6}$$

$$M(x - hu) = M(x) - hJ(x)u + O(h^2). \tag{7}$$

Subtracting and dividing by $2h$ gives the symmetric finite-difference approximation:

$$\frac{M(x+hu) - M(x-hu)}{2h} = J(x)u + O(h^2).$$

Taking the $\ell_2$-norm,

$$\left\| \frac{M(x+hu) - M(x-hu)}{2h} \right\|_2 = \|J(x)u + O(h^2)\|_2 = \|J(x)u\|_2 + O(h^2).$$

Finally, taking expectation over isotropic Gaussian perturbations $u \sim \mathcal{N}(0, I_n)$ yields

$$S(x) = \mathbb{E}_u \big[ \|J(x)u\|_2 \big] + O(h^2).$$

$\square$

This proposition establishes that the finite-difference sensitivity $S(x)$, computed using small isotropic Gaussian perturbations, provides an accurate approximation of the mean local effect of input changes on the output. Specifically, for sufficiently small step size $h$, the finite-difference estimate is equivalent, up to an $O(h^2)$ error, to the expected $\ell_2$-norm of the Jacobian applied to random Gaussian directions. Intuitively, this means that $S(x)$ captures the average magnitude of output variation induced by small, randomly oriented perturbations in the input space. By sampling $u$ from an isotropic Gaussian, all directions are treated equally, ensuring an unbiased and comprehensive measure of token sensitivity without requiring backpropagation.

## C  EXPERIMENTAL SETUP

### C.1  MODEL SETTINGS

We evaluate the effectiveness of *ZOO-Prune* on widely used VLMs, including LLaVA-v1.5-7B (Liu et al., 2024a)[1], LLaVA-v1.5-13B (Liu et al., 2024a)[2], and LLaVA-1.6-7B (Liu et al., 2024b)[3] (also referred to as LLaVA-NeXT-7B), and Qwen2.5-VL-7B (Bai et al., 2025)[4]. All LLaVA models adopt the CLIP (Radford et al., 2021) as the vision encoder and Vicuna (Chiang et al., 2023) as the base language model.

LLaVA-v1.5 models process images at $336 \times 336$ resolution, yielding 576 visual tokens, while LLaVA-NeXT-7B supports higher resolutions (up to $672 \times 672$), generating up to 2880 tokens and achieving a 6.0% gain at the cost of $3.5\times$ more computation. Qwen2.5-VL-7B, in contrast, utilizes a dynamic-resolution ViT encoder with window attention and is built upon the Qwen2.5-7B language model, supporting a variable number of visual tokens depending on input resolution. Across all experiments, our pruning is applied in a fully training-free and calibration-free manner.

### C.2  IMPLEMENTATION DETAILS

All experiments are conducted on $4\times$NVIDIA A6000 GPUs with a batch size of 1. *ZOO-Prune* is entirely training-free and attention-free, requiring no manual specification of layers in either the LMM or the vision encoder. Token selection is performed at the lightweight projection layer, which enables seamless integration across different VLM architectures. Sensitivities are also computed at this layer using simple perturbation-based operations, ensuring negligible computational overhead during inference.

For pruning ratios, we adopt 66.7%/77.8%/88.9% for LLaVA-v1.5 and 77.8%/88.9%/94.4% for LLaVA-NeXT-7B. In the latter case, we follow the implementation of VisionZip (Yang et al., 2025), where the model dynamically samples up to five image patches, resulting in as many as 2880 vision tokens. For example, with a pruning budget of 160 tokens, we retain 32 tokens per patch across five patches ($32 \times 5 = 160$). If fewer patches are sampled (e.g., four), the number of retained tokens is adjusted proportionally (e.g., 128 tokens for 160/2880). We applied a low-rank factorization ($k = 128$) to the MM-projector layers to further boost efficiency on LLaVA-Next, due to the large number of visual tokens. For the dynamic-resolution Qwen2.5-VL-7B, we evaluate at 10% and 20% token retention rates.

Finally, as validated in ablation, our method remains robust across different hyperparameter choices. Unless otherwise noted, we fix the perturbation hyperparameters to $m = 64$ and $h = 0.01$ for all experiments. Evaluation is performed using the `lmms-eval`[5] framework under official protocols and metrics.

### C.3  EVALUATION PROTOCOL AND BENCHMARK DATASETS

We conduct a comprehensive evaluation of *ZOO-Prune* across **nine widely adopted vision-language benchmarks**, spanning four core capabilities: *Visual Question Answering*, *Advanced*

---

[1]https://huggingface.co/liuhaotian/llava-v1.5-7B

[2]https://huggingface.co/liuhaotian/llava-v1.5-13B

[3]https://huggingface.co/liuhaotian/llava-v1.6-vicuna-7b

[4]https://huggingface.co/Qwen/Qwen2.5-VL-7B-Instruct-AWQ

[5]https://github.com/EvolvingLMMs-Lab/lmms-eval

*Multimodal Reasoning*, *Object Hallucination Evaluation*, and *Comprehensive Multimodal Assessment*. All experiments strictly follow the official evaluation protocols, metrics, and data splits of each benchmark to ensure fair and reproducible comparisons.

To facilitate a unified and interpretable comparison, we report both per-benchmark scores and a normalized average performance (**Avg.**), computed as the mean relative score across benchmarks with respect to the unpruned baseline. Depending on the benchmark, we report Accuracy (Acc), F1-score (F1), or Perception+Cognition (P+C), summarized in Table A. All evaluations of *ZOO-Prune* are performed under a single-model, zero-shot setting, without any task-specific fine-tuning.

**Visual Question Answering (VQA).** This category evaluates a model's ability to ground language understanding in visual content. Performance across all VQA benchmarks is measured by **Accuracy (Acc)**. We select four representative benchmarks covering diverse scenarios:

- **VQAv2-Test-Dev** (Goyal et al., 2017): General-purpose VQA with real-world images and open-ended questions.

- **GQA** (Hudson & Manning, 2019): Focused on compositional reasoning over scene graphs and structured images.

- **ScienceQA (IMG)** (Lu et al., 2022): Multimodal science questions requiring domain knowledge and diagram interpretation.

- **TextVQA** (Singh et al., 2019): Requires OCR capabilities to reason over text embedded within images.

**Advanced Multimodal Reasoning.** To probe deeper reasoning capacities beyond standard VQA, we evaluate on three challenging benchmarks. Performance on these benchmarks is also measured by **Accuracy (Acc)**:

- **MMBench** (Liu et al., 2024c): Assesses perception and reasoning across 20 fine-grained skill areas.

- **MMMU** (Yue et al., 2024): Requires expert-level multimodal reasoning across 30+ subjects grouped into six major disciplines (e.g., Art & Design, Science, Engineering, Medicine), often involving complex diagrams and charts

- **SeedBench** (Li et al., 2023a): Designed for evaluating multimodal large language models across diverse visually grounded question types, with an emphasis on perception, reasoning, and knowledge.

**Object Hallucination Evaluation.** To quantify the critical failure mode of object hallucination, we adopt the **POPE** (Li et al., 2023b) benchmark, which measures factuality in object recognition through binary existence questions. Performance is evaluated using the **F1-score (F1)** over object existence predictions, balancing precision and recall to reflect grounding reliability.

**Comprehensive Multimodal Assessment.** For a holistic evaluation of both perceptual and cognitive abilities across numerous sub-tasks (e.g., OCR, counting, attribute recognition), we employ the **MME** (Zhang et al., 2021) benchmark. It reports separate scores for Perception and Cognition tasks, summed to form the combined **Perception and Cognition score (P+C)**.

Table A: Summary of primary evaluation metrics for each benchmark.

| Benchmark | Primary Metric |
|---|---|
| VQAv2, GQA, ScienceQA, TextVQA | Accuracy (Acc.) |
| MMBench, MMMU, SeedBench | Accuracy (Acc.) |
| POPE | F1-score (F1) |
| MME | Perception + Cognition (P+C) |

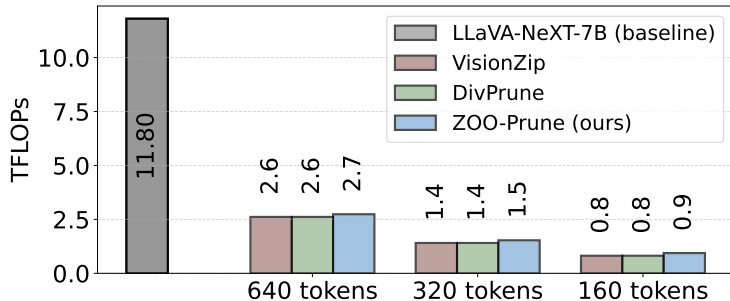

Figure A: FLOPs analysis of different token pruning methods on LLaVA-NeXT-7B across varying token retention rates. ZOO-Prune achieves similar FLOPs reduction to VisionZip and DivPrune while introducing only marginal overhead from sensitivity estimation.

## D   FURTHER INFERENCE EFFICIENCY ANALYSIS

To further investigate the computational benefits of token pruning, we analyze the floating-point operations (FLOPs) required by different methods. Figure A reports the TFLOPs measured on LLaVA-NeXT-7B with varying token retention rates (640, 320, and 160). The baseline model without pruning requires 11.8 TFLOPs, while all pruning methods substantially reduce the cost. Specifically, VisionZip and DivPrune reduce the FLOPs to 2.61, 1.41, and 0.81 TFLOPs at 640, 320, and 160 tokens, respectively. Our ZOO-Prune achieves slightly higher FLOPs due to random perturbations introduced during sensitivity estimation, with 2.74, 1.53, and 0.94 TFLOPs, respectively. Nevertheless, the overhead is negligible compared to the baseline, and ZOO-Prune achieves comparable FLOPs reduction while consistently delivering stronger accuracy.

## E   MORE VISUALIZATION

### E.1   MORE QUALITATIVE EXAMPLES

Figure B provides extensive qualitative evidence for our method's effectiveness across diverse examples, showcasing how its unified selection mechanism jointly leverages sensitivity and diversity to achieve precise token pruning. This mechanism is grounded in two concurrent principles. The zeroth-order sensitivity component identifies tokens that are most influential for answering a given question, ensuring the selection is rooted in semantic relevance. Simultaneously, the diversity criterion acts as a spatial and feature-space regularizer, guaranteeing the final token set is representative and non-redundant by preventing over-selection from visually similar regions, thereby preserving both local detail and global coverage.

The power of this dual-criterion approach is first demonstrated on questions that require fine-grained verification. For these tasks, such as confirming a person's attributes or determining a relative spatial position like the catcher's position, sensitivity is crucial for isolating the exact visual evidence. At the same time, diversity ensures that the context surrounding these key details is not entirely discarded, supporting a more robust and well-grounded verification, with correct predictions maintained even with a minimal number of retained tokens.

The method proves equally adept at handling open-ended identification tasks. Whether the challenge is to identify an object based on its relationship to others, like the car and laptop, or to recognize a subject in a scene, like the kite and deer, the fusion between the two criteria is vital. Sensitivity highlights the object of interest, while diversity provides a comprehensive, non-redundant set of its features, enabling the correct prediction. Collectively, the examples presented affirm that our sensitivity-aware diversity selection provides a principled way to maintain the visual context required for a wide spectrum of reasoning tasks.

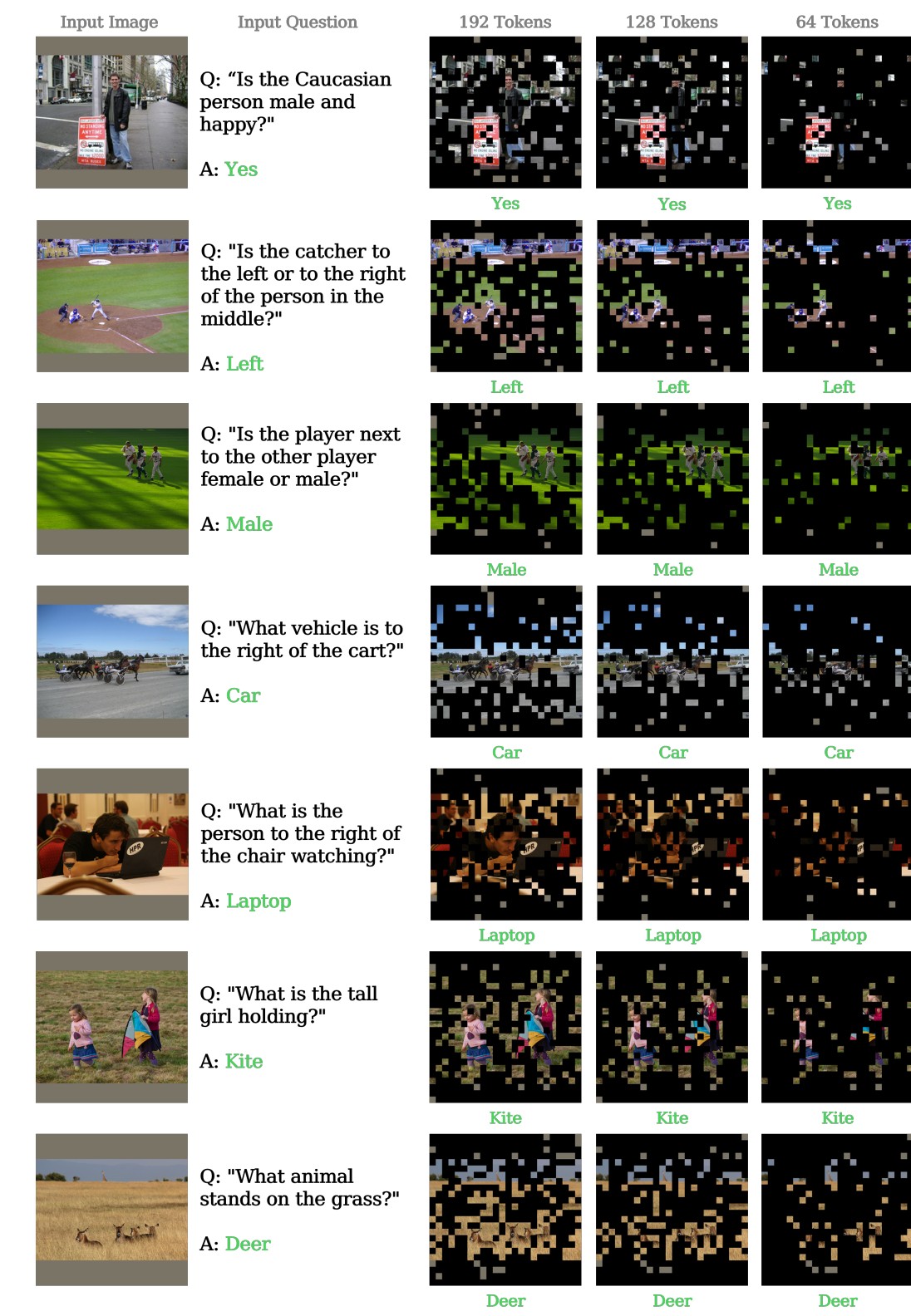

Figure B: Success cases illustrate that *ZOO-Prune* preserves key visual cues across attributes, spatial relations, and object identification, enabling accurate predictions even under aggressive pruning.

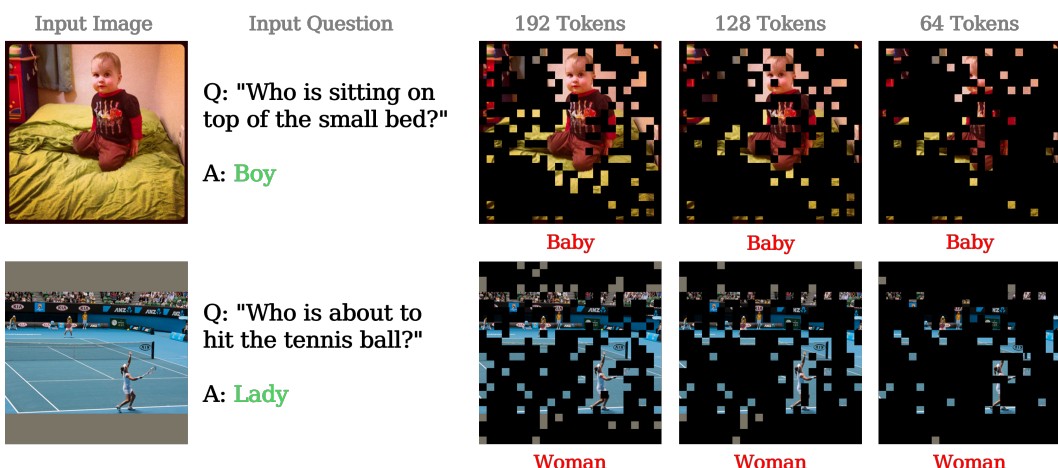

Figure C: Failure cases of *ZOO-Prune*, where predictions remain close in meaning to the reference answers yet are not exact matches. The examples illustrate typical situations where fine-grained distinctions, such as age or gender terms, lead to mismatches despite overall semantic proximity.

### E.2 FAILURE CASE EXAMPLES

Figure C illustrates representative failure cases that, while penalized by the exact-match metric, simultaneously offer strong evidence for the effectiveness of our pruning strategy. The model's predictions, such as "Baby" for "Boy" or "Woman" for "Lady," are semantically correct and align with human interpretation, yet are deemed incorrect due to word-level differences. This exposes a limitation of evaluation protocols that reward surface-level string matching over semantic equivalence. Crucially, these predictions do not signal a failure in comprehension. Instead, they show that our sensitivity-aware pruning method successfully maintains the core semantics of the image even under aggressive compression. The model's ability to generate plausible, near-synonymous concepts with only 64 tokens highlights the richness of the preserved representation and its support for high-level reasoning about key attributes like age and gender.

## F THE USE OF LARGE LANGUAGE MODELS (LLMS)

Throughout the preparation of this manuscript, we utilized Large Language Models (LLMs), specifically Google's Gemini and OpenAI's GPT-4, as writing assistance tools. Their primary role was to enhance the clarity, precision, and overall readability of the text. This included refining sentence structures, improving word choice, and ensuring grammatical correctness. In addition, the LLMs were occasionally employed to suggest alternative phrasings and to harmonize stylistic consistency across sections, thereby improving the presentation quality of the manuscript. All core research ideas, problem formulations, methodologies, experimental designs, results, and conclusions were entirely conceived, developed, and validated by the human authors. The LLMs were not involved in the generation of research content, analysis, or interpretation of findings. Instead, they served strictly as sophisticated language-polishing tools. The final version of the manuscript was carefully reviewed, edited, and approved by the authors to guarantee that it faithfully represents the original scientific contributions.

