# OpenReview forum: "Training-Free Token Pruning via Zeroth-Order Gradient Estimation in Vision-Language Models"
_ICLR.cc/2026/Conference — ICLR 2026 Conference Withdrawn Submission_

### Official Review · Reviewer_knnF · 2025-10-21

**Soundness:** 1
**Presentation:** 2
**Contribution:** 2
**Rating:** 2
**Confidence:** 4

**Summary:**

This paper proposes ZOO-Prune, a training-free token pruning framework for Vision-Language Models (VLMs) to address the high inference cost caused by redundant visual tokens. It leverages zeroth-order gradient estimation at the lightweight projection layer to measure token sensitivity and integrates this with sensitivity-aware diversity selection to ensure pruned tokens are both informative and complementary. Extensive experiments across multiple VLMs (e.g., LLaVA series, Qwen2.5-VL-7B) and benchmarks demonstrate the effectiveness of ZOO-Prune.

**Strengths:**

1. The writing is clear.

2. It is novel to use ZerOth-Order gradient estimation to estimate token importance.

**Weaknesses:**

1. **Vague concept**. The key concept _token sensitivity_ is not explained well in the Introduction. What is _token sensitivity_? Why is the _token sensitivity_ important? The authors need to provide a clear description about this key concept before using it.

2. **Lacked comparison**: There has already been man works on balancing importance and diversity, _e.g._ [1,2]. The paper lacks the conceptual and experimental comparison with these works.

[1] Ju, Chen, et al. "Turbo: Informativity-driven acceleration plug-in for vision-language large models." European Conference on Computer Vision. Cham: Springer Nature Switzerland, 2024.

[2] Yang, Longrong, et al. "Libra-Merging: Importance-redundancy and Pruning-merging Trade-off for Acceleration Plug-in in Large Vision-Language Model." Proceedings of the Computer Vision and Pattern Recognition Conference. 2025.

3. **Lacked experiments**:

(1) A main modification of the paper is to replace importance with sensitivity. However, a clear comparison between using importance and using sensitivity is lacked.

(2) The authors claim "a naive application of zeroth-order estimation would require additional forward passes through the vision encoder" in Lines 83-84. However, the experimental evidence about “significant computational overhead” is lacked.


Overall, I believe this paper has two major issues: (1) Lacked comparison. There is already considerable existing research addressing the trade-off between importance and diversity, yet the paper fails to cite or discuss these works. (2) Experimental validation. The paper lacks thorough ablation studies to justify key technical choices. For instance, why sensitivity is preferred over importance. For these reasons, I find the novelty and significance of this work insufficient to meet the acceptance standard, and therefore recommend rejection.

**Questions:**

Please refer to Weaknesses.

In addition, LLaVA-NeXT should use the dynamic resolution technique. How exactly is the token count computed in Table 5?

---

### Official Review · Reviewer_miWU · 2025-10-26

**Soundness:** 3
**Presentation:** 3
**Contribution:** 3
**Rating:** 6
**Confidence:** 4

**Summary:**

This paper introduces ZOO-Prune, a novel training-free and attention-free framework for visual token pruning in Vision-Language Models (VLMs). The core problem addressed is the high inference cost caused by a large number of redundant visual tokens. The authors propose a key intuition: tokens that are more sensitive to perturbations are more influential to the model's output. To efficiently measure this, they employ a zeroth-order (ZO) gradient estimation technique. Crucially, this estimation is performed on the lightweight projection layer rather than the entire vision encoder, serving as an effective and computationally cheap proxy for token importance. This sensitivity score is then combined with a diversity metric (inspired by DivPrune) to select a token subset that is both informative and non-redundant. The paper presents extensive experiments on multiple VLMs (LLaVA-1.5, LLaVA-NeXT, Qwen2.5-VL) and a wide array of benchmarks, demonstrating that ZOO-Prune consistently outperforms state-of-the-art training-free pruning methods, especially under aggressive pruning ratios.

**Strengths:**

- The primary strength of this paper is the novel application of zeroth-order (ZO) gradient estimation for token pruning. The idea of using the lightweight projection layer as a proxy to estimate token sensitivity is clever, well-motivated, and empirically validated (Fig. 2). This avoids costly backpropagation or full end-to-end forward passes, making the approach practical and efficient.
- The proposed framework, which combines token sensitivity with feature diversity, is principled and effective. The "Sensitivity-Aware Diversity Selection" method tackles the limitations of both sensitivity-only (potential redundancy) and diversity-only (potential loss of critical tokens) approaches. The ablation study in Table 5 provides strong evidence that both components are essential and their multiplicative fusion works best.
- The experimental setup is outstanding. The authors evaluate their method on multiple modern VLM architectures and across nine diverse and challenging benchmarks. The analysis is multifaceted, including quantitative comparisons at various pruning ratios, ablation studies, hyperparameter sensitivity analysis (Fig. 4), real-world latency measurements (Fig. 5), and insightful qualitative visualizations (Fig. 6). This level of rigor strongly supports the paper's claims.
- The paper is exceptionally well-written and easy to follow. The figures are clear, informative, and effectively convey the core concepts. Figure 1 provides an excellent motivation by visually contrasting the proposed method with prior art. Figure 3 offers a clear and concise overview of the entire ZOO-Prune pipeline.

**Weaknesses:**

- While the paper correctly argues that performing ZO estimation on the projection layer is much cheaper than on the full encoder, this step still introduces a non-trivial overhead compared to competing methods. The method requires m perturbations, resulting in 2m forward passes through the projection layer. Competing methods like DivPrune or attention-based approaches typically require only a single forward pass to obtain their scores. The FLOPs analysis in Appendix D confirms that ZOO-Prune has slightly higher FLOPs than VisionZip and DivPrune. This trade-off (slightly more computation for the pruning decision in exchange for better final accuracy) could be discussed more explicitly in the main paper.
-  While ZOO-Prune consistently shows superior performance, the improvement margin over the strongest baseline (DivPrune) can be modest in less aggressive pruning settings (e.g., in Table 1, for 192 retained tokens, the average relative accuracy is 98.6% vs. 98.0%). It would be beneficial to highlight more clearly the scenarios where the added complexity of sensitivity estimation provides the most significant benefits (which appears to be at very high pruning ratios).
- The authors demonstrate robustness to hyperparameters m and h in Figure 4, which is a strength. However, the method is not entirely hyperparameter-free. It would be helpful to briefly explain the rationale behind choosing m=64 and h=0.01 for all experiments. Was this based on a held-out set, or is it a generally applicable default?
- Lack of comparison with some key baselines [1-4]

[1] Arif, Kazi Hasan Ibn, et al. "HiRED: Attention-Guided Token Dropping for Efficient Inference of High-Resolution Vision-Language Models." Proceedings of the AAAI Conference on Artificial Intelligence. Vol. 39. No. 2. 2025. \
[2] Xing, Long, et al. "Pyramiddrop: Accelerating your large vision-language models via pyramid visual redundancy reduction." arXiv preprint arXiv:2410.17247 (2024). \
[3] Wen, Zichen, et al. "Stop looking for important tokens in multimodal language models: Duplication matters more." arXiv preprint arXiv:2502.11494 (2025). \
[4] Ye, Weihao, et al. "Fit and prune: Fast and training-free visual token pruning for multi-modal large language models." Proceedings of the AAAI Conference on Artificial Intelligence. Vol. 39. No. 21. 2025.

**Questions:**

see weaknesses

---

### Official Review · Reviewer_tKDJ · 2025-10-28

**Soundness:** 3
**Presentation:** 3
**Contribution:** 3
**Rating:** 6
**Confidence:** 4

**Summary:**

This paper proposes a training free token pruning approach tailored for Large Multimodal Models (LMMs). The key idea is to estimate token sensitivity and similarity after the projection module, and prune accordingly. At the core of the method lies a hybrid scoring function that integrates token sensitivity and token diversity. Sensitivity is approximated via a zeroth order gradient estimation achieved by applying Gaussian perturbations to the output tokens of the visual encoder, a strategy supported by theoretical justification. The diversity score is defined as 1-max("cosine similarity"), ensuring that the selected token subset contains minimal redundancy. Extensive experiments conducted on benchmark datasets such as GQA, MMB, and MMMU demonstrate an excellent accuracy–efficiency trade off, outperforming strong baselines including VisionZip and DivPrune.

**Strengths:**

This work demonstrates considerable originality by introducing an innovative approach that employs gradient information as a measure of token sensitivity and integrates it with a diversity objective, resulting in a novel and effective hybrid pruning strategy. The proposed zeroth order gradient estimation provides a practical approximation method for scenarios where model parameters are unknown. The paper exhibits strong quality. The experimental evaluation is rigorous, encompassing comparisons with state of the art methods, comprehensive ablation studies, analyses of hyperparameters such as perturbation count and scale, as well as evaluations across multiple datasets and metrics. The presentation is clear, with a well structured narrative and precise methodological exposition. Overall, the study offers evident practical significance, delivering a training free solution for accelerating large multimodal models. It achieves substantial speed ups with minimal performance degradation, making it directly valuable for real world deployment.

**Weaknesses:**

The paper could be further strengthened in several aspects: (1) The zeroth order gradient estimation method is not evaluated quantitatively against direct sensitivity computation via backpropagation; a comparison in terms of consistency and performance impact would offer stronger empirical support for adopting this technique. (2) The justification for employing gradient estimation appears weak. In the current training free token pruning setting, there seems to be no inherent barrier to computing exact gradients, as the method even assumes access to the output of the projection layer for a given input. This assumption undermines the claimed necessity of using gradient free estimation, making the motivation appear inconsistent and unconvincing.

**Questions:**

Besides the points already mentioned in the Weaknesses, I have the following additional questions and suggestions for improvement:
1. Gradient based vs. Attention based importance metrics
Could you add experiments comparing: (a) Sensitivity only vs. Attention only pruning, and (b) Sensitivity + Diversity vs. Attention + Diversity strategies? This would help evaluate the proposed gradient-based sensitivity metric's advantages over commonly used attention-based metrics.
2. Sensitivity only vs. Diversity only pruning visualization
Could you provide comparative visualizations for pruning using only sensitivity scores and only diversity scores? This would help illustrate their individual effects and the benefit of combining them.

---

### Note · Authors · 2025-11-13

I have read and agree with the venue's withdrawal policy on behalf of myself and my co-authors.